# Whole-Genome Resequencing of Ujimqin Sheep Identifies Genes Associated with Vertebral Number

**DOI:** 10.3390/ani14050677

**Published:** 2024-02-21

**Authors:** Chuanqing Zhou, Yue Zhang, Teng Ma, Dabala Wu, Yanyan Yang, Daqing Wang, Xiunan Li, Shuchun Guo, Siqi Yang, Yongli Song, Yong Zhang, Yongchun Zuo, Guifang Cao

**Affiliations:** 1College of Veterinary Medicine, Inner Mongolia Agricultural University, Hohhot 010010, China18004809342@163.com (Y.Z.);; 2State Key Laboratory of Reproductive Regulation and Breeding of Grassland Livestock, College of Life Sciences, Inner Mongolia University, Hohhot 010020, Chinazhy1956@263.net (Y.Z.); 3Inner Mongolia Academy of Agricultural and Animal Husbandry Sciences, Hohhot 010070, China; 4Key Laboratory of Animal Biotechnology of the Ministry of Agriculture, College of Veterinary Medicine, Northwest A&F University, Yangling, Xianyang 712100, China

**Keywords:** Ujimqin sheep, vertebral number, whole-genome resequencing, *ABCD4* gene

## Abstract

**Simple Summary:**

The number of vertebrae is an essential economic trait that can impact the carcass length and meat production in animals. However, our understanding of the candidate genes responsible for vertebral traits in sheep remains limited. To identify these candidate genes, we conducted a series of bioinformatics analyses. Remarkably, we discovered that the region controlling the sheep’s vertebral variation may be situated on chromosome 7. Within this region, the *ABCD4* gene emerged as a promising candidate gene for vertebral variation. This finding provides a promising perspective for unraveling the genetic basis of the vertebral number in sheep.

**Abstract:**

The number of vertebrae is a crucial economic trait that can significantly impact the carcass length and meat production in animals. However, our understanding of the quantitative trait loci (QTLs) and candidate genes associated with the vertebral number in sheep (*Ovis aries*) remains limited. To identify these candidate genes and QTLs, we collected 73 Ujimqin sheep with increased numbers of vertebrae (T13L7, T14L6, and T14L7) and 23 sheep with normal numbers of vertebrae (T13L6). Through high-throughput genome resequencing, we obtained a total of 24,130,801 effective single-nucleotide polymorphisms (SNPs). By conducting a selective-sweep analysis, we discovered that the most significantly selective region was located on chromosome 7. Within this region, we identified several genes, including *VRTN*, *SYNDIG1L*, *LTBP2*, and *ABCD4*, known to regulate the spinal development and morphology. Further, a genome-wide association study (GWAS) performed on sheep with increased and normal vertebral numbers confirmed that *ABCD4* is a candidate gene for determining the number of vertebrae in sheep. Additionally, the most significant SNP on chromosome 7 was identified as a candidate QTL. Moreover, we detected two missense mutations in the *ABCD4* gene; one of these mutations (Chr7: 89393414, C > T) at position 22 leads to the conversion of arginine (Arg) to glutamine (Gln), which is expected to negatively affect the protein’s function. Notably, a transcriptome expression profile in mouse embryonic development revealed that *ABCD4* is highly expressed during the critical period of vertebral formation (4.5–7.5 days). Our study highlights *ABCD4* as a potential major gene influencing the number of vertebrae in Ujimqin sheep, with promising prospects for future genome-assisted breeding improvements in sheep.

## 1. Introduction

In mammals, the spinal column consists of a series of repetitive bones called vertebrae. These vertebrae are categorized into different regions, including the cervical (C), thoracic (T), lumbar (L), sacral (S), and caudal (Cd) regions, based on their morphology and function along the head-to-tail axis [1]. The number and morphology of vertebrae are typically conserved within a particular species [2]. However, variations in the number of thoracic, lumbar, and thoracolumbar (the sum of the thoracic and lumbar) vertebrae have been observed in several mammalian species, such as mice [3], pigs [4], and sheep [5]. Furthermore, the number of vertebrae is associated with the body length and carcass traits, making it an important economic trait in livestock. Numerous studies have demonstrated that animals with a multi-vertebral phenotype can significantly increase the carcass length, enhance the individual meat production, and improve various production performance traits [6,7,8].

Researchers worldwide have extensively investigated the genes and quantitative trait loci (QTLs) associated with the number of vertebrae in pigs. Numerous studies on vertebral number variations in pigs have consistently highlighted the significant role of the *VRTN* gene in promoting the development of additional thoracic vertebrae [9,10,11]. Further investigation has revealed that *VRTN* plays a crucial role in the formation of thoracic vertebrae by modulating somite segmentation through the Notch signaling pathway in mice [12]. Additionally, two candidate quantitative trait nucleotides (QTNs) within the *VRTN* gene have been identified, which contribute to the observed QTL effect [13].

In recent years, there has been a growing interest among researchers in understanding the genetic factors underlying the variation in the vertebral number in sheep. Li et al. [14] conducted resequencing and selective-sweep analysis, revealing that the *VRTN* gene is a strong candidate associated with the vertebral number in Kazakh sheep. Moreover, Zhang et al. [15] investigated lamb resequencing and identified the Y87J point mutation (rs426367238) in *VRTN*, which was found to influence the number of vertebrae in Kazakh sheep. Charité et al. [16] speculated that abnormal vertebral development could be attributed to higher levels of ectopically expressed *HOXb8*. Furthermore, independent studies by Zhong et al. [17] and Li et al. [18] implicated the *SYNDIG1L* gene and *SFRP4* gene, respectively, in influencing the thoracic vertebral number and rib number. However, it is worth noting that the genes and quantitative trait loci (QTLs) associated with the number of vertebrae in Ujimqin sheep have not been reported yet. Further research is required to explore the genetic factors underlying vertebral variation, specifically in Ujimqin sheep. In addition, there have been studies on other species as well. For example, Sun et al. [19] conducted a genome-wide association analysis and found that genes *NLGN1*, *DCC*, *SLC26A7*, *TOX*, *WNT7A*, *LOC123286078*, and *LOC123280142* may be associated with the number of thoracic vertebrae in donkeys, while genes *GABBR2*, *FBXO4*, *LOC123277146*, *LOC123277359*, *BMP7*, *B3GAT1*, *EML2*, and *LRP5* may be associated with the number of lumbar vertebrae in donkeys.

Ujimqin sheep, an exceptional local breed in Inner Mongolia, China, possess remarkable characteristics, such as a large body size, high reproductive rate, strong adaptability, and abundant meat production. Consequently, owing to their larger size, Ujimqin sheep have gained popularity among local herders and individuals who prefer sheep with increased numbers of vertebrae. Despite its economic importance, the genetic information regarding the vertebral number in sheep remains limited. To address this knowledge gap, we conducted a comprehensive investigation focusing on the reproductive performance of Ujimqin sheep with multi-vertebral traits through GWAS analysis and selective-sweep analysis based on whole-genome resequencing to identify potential major genes associated with the variation in the vertebral number. Our study not only identified the key genes responsible for controlling the number of vertebrae in sheep but also expanded the genomic data available for this species. This research holds significant implications for improving the productivity of Ujimqin sheep through gene or marker selection, while deepening our understanding of the factors influencing the vertebral number.

## 2. Materials and Methods

### 2.1. Animals and Phenotypic Data

In September 2020, a total of 576 adult sheep were provided by the Ujimqin Sheep Breeding Farm in Xilingol League, Inner Mongolia. Digital radiography (DR) images were utilized to collect the phenotypic records for the vertebral number. Blood samples were collected from 96 adult Ujimqin sheep, including 23 normal sheep (T13L6) and 73 sheep with the multi-vertebral trait. The venipuncture procedure was performed under the supervision of qualified veterinarians, and it involved collecting samples from 23 sheep with 20 vertebrae (T13L7), 26 sheep with 20 vertebrae (T14L6), and 24 sheep with 21 vertebrae (T14L7). The samples were promptly transported to the laboratory and stored in a −20 °C refrigerator until the DNA extraction. The F1 generation consisted of 3 rams and 265 ewes, which were subsequently bred to produce 252 F2 lambs in April 2021. The vertebral numbers of these F2 individuals were also recorded using digital radiography (DR, MIKASA8015, Hiroshima, Japan).

### 2.2. Sequencing

Genomic DNA was extracted following the standard phenol–chloroform extraction procedure. For genomic sequencing, at least 0.5 μg of genomic DNA from each sample was used to construct a library with an insert size of ~350 bp. Paired-end sequencing libraries were constructed according to the manufacturer’s instructions (Illumina Inc., San Diego, CA, USA) and sequenced on the Illumina HiSeq X Ten Sequencer (Illumina Inc.).

### 2.3. Quality Control and Read Mapping

To ensure the reliability of the subsequent analyses, we run Trim v. 0.4 [20] to remove low-quality and adapter-contaminated reads, resulting in clean data. To detect genetic variations in the sheep genomes, we utilized BWA software v.0.7 [21] to map the clean reads to the reference genome, followed by the identification of polymorphic sites using SAMtools software v.1.19 [22]. Subsequently, we utilized BWA software to index the sequencing data (-index) and generate sai files (bwa aln -c -t 3 -f treads.sai reference.fa treads.fastq). The SA coordinates were then converted to a SAM file (bwa samse -f single.sam reference.fa single.sai single.fastq). SAMtools software v.1.19 was employed to convert the SAM file to a BAM file (samtools view -bS example.sam -o example.bam). SNP calling was conducted using bcftools v. 1.3.1 (bcftools mpileup -Ou file.bam -f reference.fna|bcftools call -mv -o result.vcf). For the SNP quality control, VCFtools software v.0.1.16 was utilized with the following parameters: the Hardy–Weinberg equilibrium was <0.0001 and the minor allele frequency was <0.05. The reference genome employed throughout these procedures was Oar_rambouillet_v1.0. 

### 2.4. Statistical Analysis

To identify potential signals of selection that occurred during the breeding of sheep with a higher number of vertebrae, we employed Plink software v.1.9 [23] to calculate the Fst value and Pi value for variations observed across the entire genome. The analysis was performed using a window size of 50 kb and a step size of 5 kb. However, the calculation of the Pi value required a sample ratio of 1:1. Thus, we randomly selected 7 sheep with T13L7, 8 sheep with T14L6, and 8 sheep with T14L7 (The sample list is provided in Appendix A). The number of sheep with T13L6 remained at 23. Additionally, genome-wide association analysis was conducted using Plink. At the whole-chromosome level, the significance threshold for the GWAS was set at 6 (*p* < 1 × 10^−6^, using the Bonferroni correction for the significance threshold). The quantile–quantile (Q–Q) plot was utilized to visualize the distribution of the expected log10 (*p*-value) against the expansion factor (λ). Furthermore, Manhattan plots were generated to depict the correlation diagrams and highlight significant SNPs, with threshold lines indicating significance levels. The correlation analysis employed the logistic regression model integrated into Plink. The equation of the model is usually as follows:Logit (P) = β_0_ + β_1_ X_1_ + β_2_ X_2_ + … + β_n_ X_n_

Among them, logit (P) is the logarithmic probability of the phenotypic variation; P is the probability of the phenotypic variation; β_0_ is the intercept; β_1_, β_2_, …, β_n_ are the regression coefficients of the model; and X_1_, X_2_, …, and X_n_ are the explanatory variables (genotypes).

Both the Manhattan plot and Q–Q plot were created using CMplot v. 4.5.1 [24]. To annotate the candidate SNPs, we utilized the ANNOVAR package v. 2021 [25] with the *O. aries* reference genome (Oar_rambouillet_v1.0). In addition, the method for adjusting *p*-values is FDR in the Gene Ontology (GO) enrichment analysis.

### 2.5. RNA-Seq Analysis

We obtained transcriptome sequencing data of the early embryonic development in mice (GSE121650 and GSE87038) for the RNA-Seq analysis. The number of samples is not entirely consistent for each time point. Specifically, there are 4 samples for the 2-cell stage, 4-cell stage, blastocyst stage, and morula stage. For embryonic development, there are 5 samples for 4.5 days, 6.5 days, and 8.5 days; 6 samples for 5.5 days and 7.5 days; 3 samples for the 8-cell stage; 2 samples for 9.5 days; and 1 sample each for 10.5 days and 11.5 days. Clean reads were obtained by removing reads containing the adapter, reads containing ploy-N, and low-quality reads from the raw data by calculating the Q20, Q30, and GC content of the clean reads, respectively. We used HISAT v. 2.1.0 [26] and StringTie v. 2.0 [27] to map the paired-end reads to the mouse reference genome (GRCm39) and assemble the reads, respectively (hisat2 -p 10 -x reference -1 ${id}_1_val_1.fq.gz -2 ${id}_2_val_2.fq.gz -S ${id}.hisat.sam). The number of reads matched to an expressed gene was standardized as transcripts per kilobase million (TPM) values. The expression of the *ABCD4* gene was drawn using R software v.4.2.3.

### 2.6. Genotyping Causal Variants of the ABCD4 Candidate Gene

SNP1 (Chr7:89387652, G > A); SNP2 (Chr7:89384139, G > C); and SNP3 (Chr7:89393414, C > T) were identified as candidate causal mutations for the number of vertebrae; therefore, we detected the *ABCD4* locus in F2 individuals to investigate this trait. The set of primers that we used to amplify the genomic DNA is given in Appendix A. The PCR amplification was performed with 1.5 mmol of MgCl_2_ and optimized annealing temperatures according to standard methods. All the PCR products were unidirectionally sequenced with the original PCR primers (Appendix A) in a 3130XL genetic analyzer (Applied Biosystems, Waltham, MA, USA).

## 3. Results

### 3.1. Investigation and Evaluation of Vertebral Number in Ujimqin Sheep

In general, sheep have 7 cervical vertebrae (C7), 13 thoracic vertebrae (T13), 6 lumbar vertebrae (L6), 4 sacral vertebrae (S4), and a variable number of caudal vertebrae. In this study, we investigated the number of vertebrae (thoracic and lumbar) in 576 adult sheep and 525 lambs. We observed four types of vertebral numbers, namely, T13L6, T13L7, T14L6, and T14L7 (Figure 1A–D). Among these types, T13L6 is considered as the primitive form of vertebrae in mammals [2]. The proportions of these vertebral types were as follows: T13L6 (3.6%), T13L7 (56.8%), T14L6 (33.3%), and T14L7 (6.3%) (Figure 1, Table 1). Notably, a significant majority (96.4%) of the Ujimqin sheep displayed multi-vertebral traits (Figure 1E, Table 1).

To further assess the reproductive performance of the multi-vertebral sheep, we examined the number of vertebrae in 252 lambs produced by 3 rams with 21 vertebrae (T14L7) and 265 ewes. Artificial insemination was employed for mating purposes. Our findings revealed that a significant proportion (99.2%) of the offspring displayed multi-vertebral traits, while the number of offspring with the common vertebral form (T13L6) was much lower (Table 2). 

### 3.2. Whole-Genome Resequencing and SNP Identification

To identify candidate genes associated with increased numbers of vertebrae in sheep, we collected 23 samples with the T13L6 trait, 24 samples with the T13L7 trait, 26 samples with the T14L6 trait, and 23 samples with the T14L7 trait. Whole-genome high-throughput sequencing was performed; the sequencing depth was 10×, and the average data size was 28.46 Gb (Appendix A). The mapping rate of the clean reads to the sheep reference genome was approximately 99.83% (Appendix A). And we initially obtained 40,306,349 unfiltered SNPs by comparing the reads with the reference genome; the genotypic rate was 97%, and the SNP call rate was 1.5%. After filtering, we identified a total of 24,130,801 effective single-nucleotide polymorphisms (SNPs) for the analysis.

### 3.3. Selective-Sweep Analysis

To identify regions of the sheep genome that have undergone a recent strong selection, we examined areas exhibiting reduced nucleotide diversity (Pi) and increased genetic distance (Fst). The analysis of the Pi ratio revealed a putative selected genomic region located in the 89.25–90.00 Mb region of chromosome 7 (Figure 2A). Additionally, the analysis of the Fst values identified the most significant region in the 89.05–89.55 Mb region of chromosome 7 (Figure 2C, Table 3), which overlapped with the Pi results. We annotated this overlapping region (Chr7: 89.25–89.55 Mb) and identified 8 genes within it. Notably, the region encompasses the *VRTN*, *SYNDIG1L*, *LTBP2*, and *ABCD4* genes, which are known to be associated with development and morphology (Figure 2A). Consequently, the joint analysis of Pi and Fst led us to hypothesize that this region may be a potential candidate area involved in regulating the number of lumbar vertebrae in sheep. Furthermore, the presence of gene clusters within this region suggests that these genes may collectively contribute to the formation of the vertebrae.

### 3.4. Genome-Wide Association Analysis

A genome-wide association study (GWAS) was conducted to further identify genes and quantitative trait loci (QTLs) associated with the number of vertebrae in Ujimqin sheep. The GWAS results revealed that a total of 64 SNPs showed a strong correlation with the number of vertebrae, surpassing the significance threshold of 6 (*p* < 10^−6^) (Figure 3A,C). The most significant SNPs, ranking in the top 3, were located on SSC6 and SSC7 (Appendix A). However, the two SNPs on SSC6 were situated in an intergenic region without any nearby genes within a 1 Mb span (Chr6: 79.566288–81.566373 Mb). In contrast, an SNP (SNP1, Chr7:89387652, G > A) was identified within an intron of the *ABCD4* gene on chromosome 7 (Figure 3B). Notably, all the SNPs surpassing the threshold on chromosome 7 exhibited linkage disequilibrium (Figure 3D). These findings indicate that *ABCD4*, along with SNP1 (Chr7:89387652, G > A), represents a robust candidate gene and a QTL associated with the number of vertebrae in Ujimqin sheep, respectively.

### 3.5. Mutation Analysis

The analysis of the base mutations in the *ABCD4* gene (Chr7:89379307…89394704) revealed a higher frequency of C > T and G > A mutations (Figure 4A,B, respectively). These mutations may contribute to a decrease in the GC content within this specific region of the chromosome. It is well-known that the GC content can impact the DNA density, subsequently influencing the stability of genes to some extent. Upon annotating the SNPs within the *ABCD4* gene, we identified SNP2 (Chr7:89384139, G > C) and SNP3 (Chr7:89393414, C > T) as missense mutations (Figure 4A,B, respectively). These mutations are located in exon 8 and exon 18 of the *ABCD4* gene, respectively (Figure 2B).

The protein structure of *ABCD4* in sheep was predicted using the Alpha-Fold database [28], revealing that the main structure of the *ABCD4* protein comprises six transmembrane helices (TMHs) [29] (Figure 4C). SNP2 (Chr7:89384139, G > C) is located at amino acid position 375 in the loop region connecting the fourth and fifth TMHs. This SNP causes a change from G to C at this position, resulting in the conversion from proline (Pro) to arginine (Arg). However, this change may not significantly affect the biological function of the protein. On the other hand, SNP3 (Chr7:89393414, C > T) is located at amino acid position 22, just before entering the first transmembrane helix. This SNP leads to a change from C to T at this position, resulting in the conversion from arginine (Arg) at position 22 to glutamine (Gln). This change significantly alters the properties of the amino acid, affecting the interactions with surrounding amino acids and the protein’s overall structure. This alteration may influence the substrate-binding capability of *ABCD4*, thereby affecting the function and regulatory mechanism of the protein (Figure 4C).

### 3.6. SNP Screening and Sequencing Validation

The most significant SNP was obtained from GWAS (SNP1, Chr7:89387652, G > A) on SSC7, and the two missense mutations (SNP2, Chr7:89384139, G > C, and SNP3, Chr7:89393414, C > T) were sequenced to further investigate their association with the number of vertebrae in Ujimqin sheep. The results showed that SNP2 (Chr7:89384139, G > C) was not observed in the sequenced individuals. However, SNP1 (Chr7:89387652, G > A) and SNP3 (Chr7:89393414, C > T) were found exclusively in the sequenced sheep with the multi-vertebral trait (Figure 5). This suggests that SNP1 (Chr7:89387652, G > A) and SNP3 (Chr7:89393414, C > T) are causal mutations influencing the formation of the vertebrae. On the other hand, SNP2 (Chr7:89384139, G > C) is likely to be present in only a small number of sheep with the multi-vertebral trait and may not be a strong candidate mutation for the variation in the number of vertebrae.

### 3.7. The Expression of ABCD4 and Related Gene Enrichment Analysis

To investigate the functional mechanism of *ABCD4*, we utilized GEPIA2 [30] to identify the top 100 genes with expression patterns similar to that of *ABCD4* (Appendix A). The Gene Ontology enrichment analysis revealed that these genes were closely associated with mRNA splicing and the process of mRNA emptying (Figure 6A, Appendix A). Furthermore, we utilized the STRING tool [31] to obtain 40 genes that exhibited co-expression with *ABCD4*, thereby validating the results of the Gene Ontology enrichment analysis. As depicted in Figure 6C, these 40 genes displayed strong mutual correlations, and they were significantly enriched in molecular metabolic processes (Appendix A). These findings led us to speculate that *ABCD4* may play a role in the processes of transcription and molecular metabolic processes, potentially contributing to vertebral development.

The development of somites, which eventually give rise to vertebrae, initiates during the gastrula stage. We investigated the expression pattern of the *ABCD4* gene during embryonic development (from 2 cells to E11.5) in mice and obtained a transcriptome expression profile (Figure 6B). The results revealed that *ABCD4* exhibited high expression levels during the critical period for vertebral formation, specifically from E4.5 to E7.5. This finding supports the notion that *ABCD4* is a strong candidate gene for the variation in the number of vertebrae, as indicated by our study. However, further research on sheep is required to confirm this hypothesis.

## 4. Discussion

In animal husbandry, the practice for selectively breeding animals with desirable traits has been employed to enhance production. Consequently, this prolonged artificial breeding has led to a reduction in the number of polymorphisms within the species [32]. The number of vertebrae is an economically significant trait that can impact various production traits in animals. In the case of pigs, incorporating information about the rib number into the selection process has been shown to improve production traits [32]. Similarly, studies have indicated the presence of numerical variations in the rib number in sheep, and selecting sheep with a higher number of ribs may potentially improve the carcass quality, including factors like the carcass length and weight [33,34].

In sheep, the normal number of vertebrae is typically T13L6, and variations in the vertebral number are rare. However, our investigation revealed a significant presence of multi-vertebral sheep in the population we studied, which can be attributed to directed breeding practices. Local herdsmen prefer sheep with a greater number of vertebrae owing to the potential for increased productivity. Interestingly, our findings also indicate that the multi-vertebral trait displays good genetic stability, and the selective breeding of rams with multi-vertebral traits is advantageous for the production of multi-vertebral sheep and the establishment of a core population of multi-vertebral Ujimqin sheep. Although the specific genetic basis underlying the number of vertebrae remains unclear, we believe it is a stable trait that warrants further study, particularly for sheep breeding purposes. 

To identify the selective signals associated with an increased number of vertebrae in sheep, we conducted large-scale resequencing of Ujimqin sheep with T13L6, T14L6, T13L7, and T14L7 traits. We obtained 24,130,801 single-nucleotide polymorphisms (SNPs), and the number of SNPs identified was similar to that reported in a previous study [35], suggesting that the sequencing data possessed high quality and could be utilized for subsequent analyses. The selective-sweep analysis revealed a potential selected region in the sheep genome that is correlated with an increase in the number of vertebrae. Within this region, we identified genes that are known to be involved in the development and growth of vertebrae. Notably, the *VRTN* gene has been extensively studied and shown to play a significant role in the development of thoracic vertebrae in mammals [12]. The *SYNDIG1L* gene has also been implicated in determining the number of vertebrae in sheep [17], while the *LTBP2* gene has been associated with thoracic vertebral and rib numbers in pigs [36]. Additionally, the *ABCD4* gene, which is known to be involved in vitamin B12 metabolism [37], has been identified as a candidate gene significantly associated with total vertebral and rib numbers in pigs through genome-wide association studies [38,39]. Interestingly, *NPC2* genes were also found within this region, although they have not been reported to directly affect vertebral development in mammals. However, they have been shown to participate in the shaping of the zebrafish spine, where a decrease in the *NPC2* expression leads to developmental defects and body axis distortion through the Notch signaling pathway [40]. These findings validate the accuracy of our sequencing approach in identifying genomic regions associated with variations in the number of vertebrae in sheep.

By conducting a genome-wide association analysis, we have once again associated the *ABCD4* gene with the number of vertebrae in sheep. We have identified two SNPs within the *ABCD4* gene as candidate QTLs through our screening process. The missense mutation SNP3 (Chr7:89393414, C > T) leads to changes in amino acids, thereby altering the function of the protein. Although the most prominent SNP (SNP1, Chr7:89387652, G > A) obtained through the GWAS is located in an intron, numerous studies have demonstrated that intron mutations can have important impacts on the phenotype of an individual. Although introns have been traditionally considered as non-coding regions, recent research has shown that they play independent roles. For instance, some introns can promote the expression of messenger RNA (mRNA) and encode small nucleolar RNA (snoRNA) and microRNA (miRNA) that can have regulatory functions [41]. Furthermore, introns themselves have been found to have independent functions. Morgan et al. revealed that introns can mediate cellular responses to hunger, and Parenteau et al. discovered that stable introns in yeast genes regulate the yeast’s growth rate under pressure, enhancing the yeast’s adaptability [42,43]. Therefore, we speculate that the mutation in the *ABCD4* gene can, indeed, influence the number of vertebrae in sheep.

*ABCD4* is a membrane protein that functions as an ATP enzyme. Several studies have demonstrated that *ABCD4* is involved in the intracellular processing of vitamin B12. Mutations in the *ABCD4* gene can lead to metabolic errors related to vitamin B12 [37]. Vitamin B12 has been shown to stimulate the proliferation and osteogenesis of osteoblasts [44]. Additionally, the depletion of vitamin B12 can alter components of the Wnt pathway [45,46], and the Wnt and Notch signaling pathways have been shown to interact and play crucial roles in somite development through the segmentation clock [47,48,49]. Therefore, *ABCD4* should be considered as a critical candidate gene that affects the number of vertebrae.

## 5. Conclusions

In this study, we performed a comprehensive analysis to investigate the genetic factors influencing the spinal development and morphology in Ujimqin sheep with different numbers of vertebrae. Through our analysis, we identified a significant region on SSC7 associated with the number of vertebrae, and the *ABCD4* gene located within this region was found to be potentially linked to the variation in the vertebral number in sheep. Our findings provide valuable insights into the genetic mechanisms underlying the vertebral variation in sheep.

## Figures and Tables

**Figure 1 animals-14-00677-f001:**
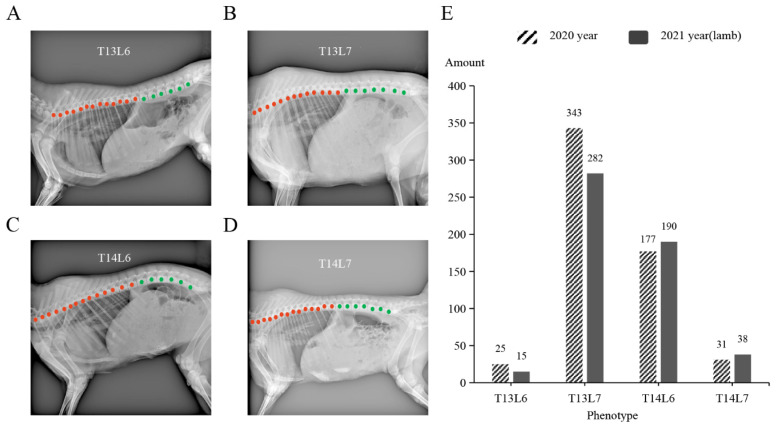
Four types of vertebral numbers in Ujimqin sheep. (**A**–**D**) show the DR images of the thoracolumbar vertebrae in Ujimqin Sheep with T13L6, T13L7, T14L6, and T14L7 traits. The red marks indicate the thoracic vertebra, and the green marks represent the lumbar vertebra. (**E**) Statistical bar chart of vertebral number in adult sheep in 2020 and lambs born in 2021. The *X*-axis represents the phenotype of the sheep, and the *Y*-axis represents the number of sheep.

**Figure 2 animals-14-00677-f002:**
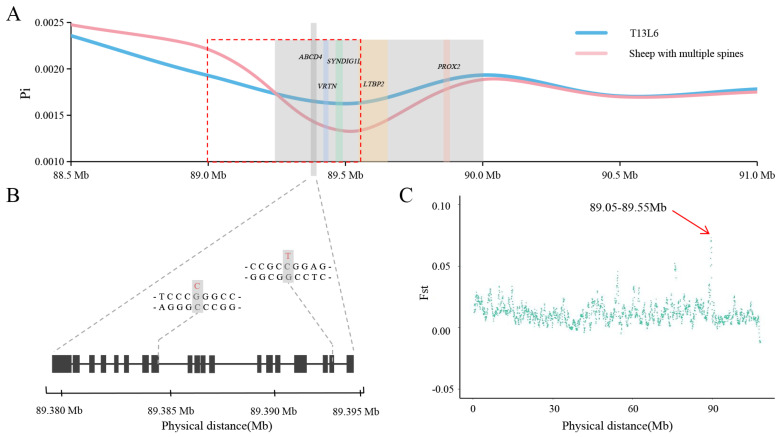
Selection analyses identified a candidate region. (**A**) Pi value in 50 kb sliding windows with 5 kb steps in T13L6 (blue line) and multi-vertebral sheep (red line) from 88.5 to 91.0 Mb on chromosome 7. The shaded region indicates 89.25–90.0 Mb screened by Pi, and the red dotted box region represents 89.0–89.55 Mb screened by Fst. The shadows of the other colors represent genes in this region. (**B**) Schematic diagram of *ABCD4* exon structure and two missense mutations in *ABCD4* exon. *X*-axis represents the physical distance on chromosome 7. (**C**) Fst distributions plotted for sheep chromosome 7.

**Figure 3 animals-14-00677-f003:**
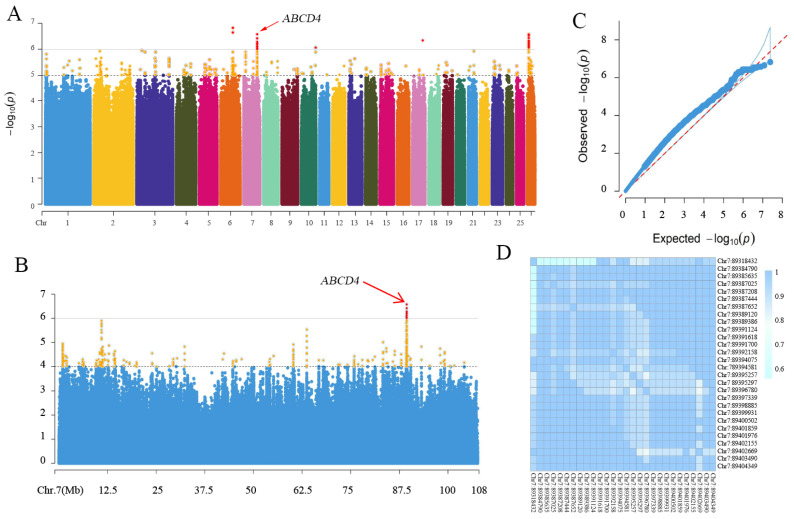
(**A**) Genome-wide association study for vertebral number in Ujimqin sheep. Manhattan plots of GWAS results. The *X*-axis shows SNPs across chromosomes from SSC1 to SSC26, and the *Y*-axis represents the −log10(*p*) value. The top line indicates the genome-wide significant thresholds (*p* < 1 × 10^−6^). (**B**) Manhattan plots for SSC7 of GWAS results. The top line indicates the genome-wide significant thresholds (*p* < 10^−6^). (**C**) Q–Q plot of GWAS analysis. (**D**) Heat map of linkage disequilibrium between significant SNPs on SSC7. The *X*-axis and *Y*-axis represent SNPs.

**Figure 4 animals-14-00677-f004:**
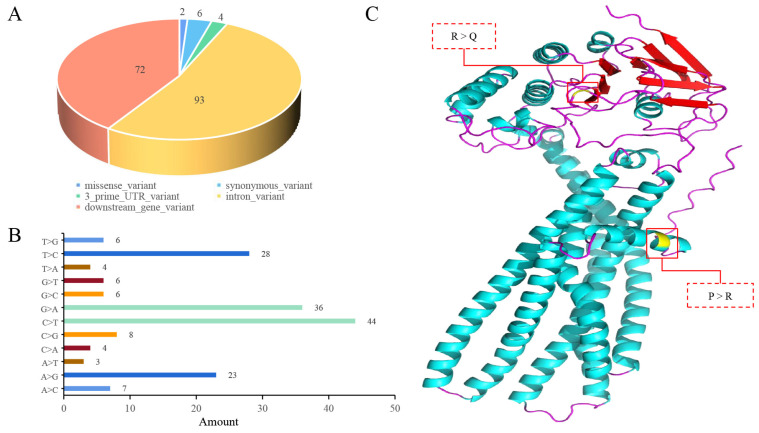
Statistics of mutation types in *ABCD4* gene (Chr7: 89379307–89394764). (**A**) Pie plot of SNP types in *ABCD4* gene (Chr7: 89379307–89394764). (**B**) Bar plot of base types in *ABCD4* gene (Chr7: 89379307–89394764). The *X*-axis represents the mutation number, and the *Y*-axis represents the mutation type. (**C**) The three-dimensional protein structure of *ABCD4* (sheep) is predicted using Alpha-Fold, and the location of the protein structure is affected by missense mutations in the *ABCD4* gene.

**Figure 5 animals-14-00677-f005:**
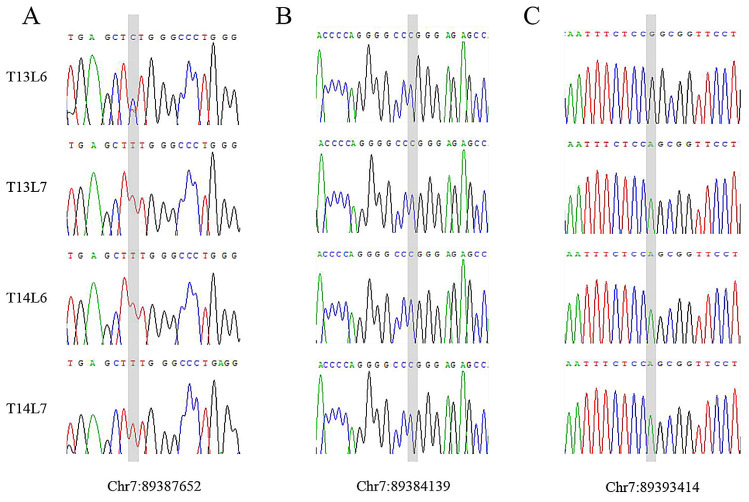
Sanger sequencing of the three candidate SNPs in the sheep *ABCD4* gene (sequencing of complementary chain). The shadows in figures (**A**–**C**) indicate the locations of candidate mutations and base types in sheep with different vertebral numbers.

**Figure 6 animals-14-00677-f006:**
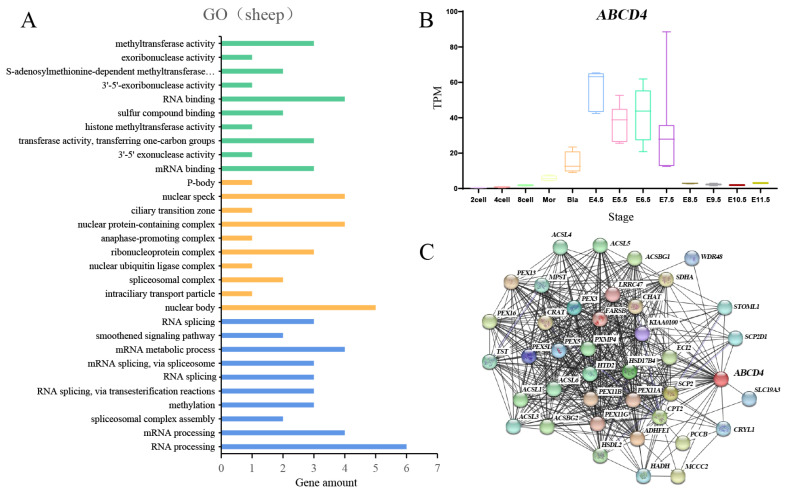
The expression of *ABCD4* and related gene enrichment analysis. (**A**) Gene Ontology (GO) analysis of the top 100 genes co-expressed with *ABCD4* obtained using GEPIA2. Green indicates the top 10 GO terms of the molecular function, yellow indicates the top 10 GO terms of the cellular component, and blue indicates the top 10 GO terms of the biological process. The *X*-axis represents the number of genes, and the *Y*-axis represents the GO term. (**B**) The expression of the *ABCD4* gene in mice during the embryonic development period (from 2 cells to E11.5). The *X*-axis represents the developmental stage, and the *Y*-axis represents TPM (transcripts per kilobase million). (**C**) Co-expression network of 40 genes co-expressed with *ABCD4* was obtained using the STRING tool.

**Table 1 animals-14-00677-t001:** Statistical table of multi-spine phenotypes in adult sheep in 2020 and lambs in 2021.

Year	Number	T13L6		T13L7		T14L6		T14L7	
		Number	Proportion	Number	Proportion	Number	Proportion	Number	Proportion
2020	576	25	4.3	343	59.6	177	30.7	31	5.4
2021	525	15	2.9	282	53.7	190	36.2	38	7.2
Total	1101	40	3.6	625	56.8	367	33.3	69	6.3

**Table 2 animals-14-00677-t002:** Phenotypic statistics of lambs produced by mating rams (T14L7) with ewes of different phenotypes.

Paternal Phenotype	Maternal Phenotype (Number)	Offspring (T13L6)		Offspring (T13L7)		Offspring (T14L6)		Offspring (T14L7)	
		Number	Proportion	Number	Proportion	Number	Proportion	Number	Proportion
T14L7	T13L6 (11)	2	18.2	8	72.7	1	9.1	-	-
T14L7	T13L7 (148)	1	0.7	91	61.5	49	33.1	7	4.7
T14L7	T14L6 (94)	1	1.1	25	26.6	56	56.9	12	12.7
T14L7	T14L7 (12)	5	41.7	5	41.7	2	16.6	-	-

**Table 3 animals-14-00677-t003:** Top 5 regions with significant Fst values on chromosome 7.

Chromosome	Bin_Start	Bin_End	Fst
7	89,050,001	89,550,000	0.0729947
7	89,100,001	89,600,000	0.0707678
7	89,150,001	89,650,000	0.0701816
7	89,000,001	89,500,000	0.0642738
7	89,200,001	89,700,000	0.0653491

## Data Availability

The public transcriptome sequencing data of the mice were downloaded using the GSE121650 and GSE87038 accession numbers from the GEO database at September https://www.ncbi.nlm.nih.gov/geo/ (accessed on 12 September 2022). The data of the genome-wide association study in Ujimqin sheep are available from the corresponding author on reasonable request.

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
