# Peer review of "Whole-Genome Resequencing of Ujimqin Sheep Identifies Genes Associated with Vertebral Number"

_animals, 2024, doi:10.3390/ani14050677_

Round 1

Reviewer 1 Report

Comments and Suggestions for Authors

Dear authors,

I will recommend your manuscript for publication, but I has some questions for edition.

In Abstract – information about aminoacid chain changes will be informative, add it

In Introduction – I recommend add recent information about vertebrae number controls in mammalian and other classes of animals

In Animals and Phenotype Data – Did you investigated meat production traits in sheep with different vertebrae numbers? Add equipment of Digital Radiography.

In Quality Control and Reads Mapping – I recommend add information about sequence coverage and average data (in Gb) per sample obtained. And what is reference genome was using for assembling?

In Figure 3 – Manhattan plot A not informative, I recommend change it like plot B. Why you are not using Bonferrony correction for significance threshold?

In Figure 5 – Not clear – you were using Sanger sequencing?

Regards,

Author Response

Reviewer: 1

Comments and Suggestions for Authors

Dear authors,

I will recommend your manuscript for publication, but I has some questions for edition.

(1) In Abstract – information about amino acid chain changes will be informative, add it

In Introduction – I recommend add recent information about vertebrae number controls in mammalian and other classes of animals

Response: Thank you very much for your comments. We have added information on amino acid chain changes in the Abstract (See Page 1, the highlighted part). And we have added recent information about vertebrae number controls in other classes of animals (See Page 2, the highlighted part).

(2) In Animals and Phenotype Data – Did you investigated meat production traits in sheep with different vertebrae numbers? Add equipment of Digital Radiography.

Response: Thank you for your constructive suggestions. I'm sorry, we haven't investigated whether there is a change in raw meat yield in sheep with different vertebrae, but the article “The number of ribs and vertebrae in a Pietrain cross: variation, heritability and effects on performance traits. Journal of Animal Breeding and Genetics, 2004 (6): 392403.” reported that in pigs, increasing the number of thoracolumbar vertebrae can significantly increase their body length and weight. For example, the 14% variation in the body length of European commercial pigs can be attributed to the variation in the number of thoracolumbar vertebrae. For each additional thoracic vertebra or lumbar vertebra, the carcass length of the pig increases by 15 mm. In addition, “Possible introgression of the VRTN mutation increasing vertebral number, carcass length and teat number from Chinese pigs into European pigs. Sci Rep, 2016, 6:19240.” also have shown that increasing the number of lumbar vertebrae of pigs can significantly increase their average daily gain and daily gain during fattening period.

The conclusion of “The meat production performance of Multi-vertebrae sheep is significantly superior to that of ordinary sheep, with an increase in live weight of 4.77-7.6 kg, carcass weight of 4.12-5.59 kg, and net meat weight of 3.36-4.90 kg.” has been reported by article titled “多脊椎蒙古羊的胸腰椎长度及其产肉性能分析. 内蒙古农牧学院学报, 1998,  19(03):1-5”.

The study of Yang Jianmei found in her article titled金川牦牛胸、腰椎组型变异及其与产肉性能的相关分析. 家畜生态学报, 2015, 36(09):26-30. that if Jinchuan yak had one more thoracic vertebra or lumbar vertebra, the average length of spine increased by about 7.60 cm, and the area of eye muscle increased by 8.29 square cm. Under the T15L5 thoracolumbar combination, the meat yield of bulls was about 17 kg higher than that of T14L5 combinations, and that of female yaks was about 5 kg.

Although we have not conducted a survey on this at present, with the support of these literatures, we believe that the yield of raw meat with multiple spines is high. Of course, we will also conduct investigations and studies in this area in the future, and will further elaborate in future studies.

In addition, the equipment of Digital Radiography is MIKASA8015(JAPAN),which has been added to our manuscript(See Page 3, the highlighted part).

(3) In Quality Control and Reads Mapping – I recommend add information about sequence coverage and average data (in Gb) per sample obtained. And what is reference genome was using for assembling?

Response: Thank you for your valuable advice. The sequence coverage of this study is 99.83%, the average data size is 8.22Gb, which have also added to the manuscript (See Page3, the highlighted part ) and the data size of each sample has been put into the supplementary data(Table S9).In addition, the reference genome we used is Oar_rambouillet_v1.0 (See Page 3, the highlighted part).

(4) In Figure 3 – Manhattan plot A not informative, I recommend change it like plot B. Why you are not using Bonferrony correction for significance threshold?

Response: Thank you again for your constructive suggestions. I have added it to the supplementary figure (Supplementary Figure). About significance threshold, considering the influence of linkage, we calculate the number of independent SNP and take the negative logarithm of 1 divided by the number of independent markers to represent the threshold. In our study, the independent mark is 96584, so the threshold is set to 6 (p-value < 0.000001). Relevant content has been added to the manuscript (See Page 3, the highlighted part).

(5) In Figure 5 – Not clear – you were using Sanger sequencing?

Response:Thank you for pointing that out. We used Sanger sequencing in the Figure 5.

Reviewer 2 Report

Comments and Suggestions for Authors

General Comments

The research explores identifying the candidate genes associated with the vertebrae number of Ujimqin sheep. Besides its practical applications, the study will make a significant contribution to understanding QTL and candidate genes linked to vertebrae number in sheep genetics, offering valuable insights for researchers and stakeholders in the livestock industry. However, there are certain areas that should be addressed to improve the manuscript’s potential for publication.

Specific comments:

Point 1: In the introduction section, include the reference to Name et al. immediately, rather than at the end of the sentence.

Point 2: It appears that the author did not take into account the SNP and genotypic call rate during quality control. Please provide an explanation.

Point 3: Please provide a brief description of the statistical model used for GWAS in the manuscript, including the corresponding equation.

Point 4: What was the rationale for using a p-value < 0.000001 to define significant SNPs?

Point 5: Why the authors did not consider any statistical test to adjust for multiple comparisons?

Point 6: The article notes an important observation regarding the QQ plots. A noticeable deviation from the expectation (red line) is clearly observed. Could you please provide the lambda value for the QQ plot?

Point 7: In the Gene Ontology, could you please provide the reasons for presenting only Molecular Function? Have you considered including Biological Process, Cellular Components, and KEGG Pathway as well?

Point 8: The conclusions section in this manuscript requires improvement to provide a clearer explanation of the study's results, its significance, and the implications for future research.

Author Response

Reviewer: 2

Comments and Suggestions for Authors

General Comments

The research explores identifying the candidate genes associated with the vertebrae number of Ujimqin sheep. Besides its practical applications, the study will make a significant contribution to understanding QTL and candidate genes linked to vertebrae number in sheep genetics, offering valuable insights for researchers and stakeholders in the livestock industry. However, there are certain areas that should be addressed to improve the manuscript’s potential for publication.

Specific comments:

Point 1: In the introduction section, include the reference to Name et al. immediately, rather than at the end of the sentence.

Response: Thank you for your professional advice. We have changed it in the introduction section (See Page 2, the highlighted part).

Point 2: It appears that the author did not take into account the SNP and genotypic call rate during quality control. Please provide an explanation.

Response: Thank you for your valuable advice. We calculated the call rate of SNP, and we initially obtained 40306349 unfiltered SNPs, of which 1209192 were successful untyped, that is, the genotypic rate was 97%, and the SNP call rate was 1.5%. Relevant explanation content has been added to the manuscript (See Page 3, the highlighted part).

Point 3: Please provide a brief description of the statistical model used for GWAS in the manuscript, including the corresponding equation.

Response: Thank you again for your constructive suggestions. We have added relevant descriptions to the manuscript (See Page 4, the highlighted part).

Point 4: What was the rationale for using a p-value < 0.000001 to define significant SNPs?

Response: Thank you for your professional advice. Considering the influence of linkage, we calculate the number of independent SNP and take the negative logarithm of 1 divided by the number of independent markers to represent the threshold. In our study, the independent marks are 96584 SNPs, so the threshold is set to 6 (p-value < 0.000001). Relevant explanation content has been added to the manuscript (See Page 3, the highlighted part)

Point 5: Why the authors did not consider any statistical test to adjust for multiple comparisons?

Response: Thank you for your helpful suggestions. We have considered the threshold determined by multiple tests (Bonferrony), and we have added it to the manuscript (See Page 3, the highlighted part).

Point 6: The article notes an important observation regarding the QQ plots. A noticeable deviation from the expectation (red line) is clearly observed. Could you please provide the lambda value for the QQ plot?

Response: Thank you for your valuable advice, the calculated lambda value is 0.89.

Point 7: In the Gene Ontology, could you please provide the reasons for presenting only Molecular Function? Have you considered including Biological Process, Cellular Components, and KEGG Pathway as well?

Response:Thank you for pointing that out. We have added the full GO function enrichment to the Figure 6A, including molecular function, biological processes and cellular components (See Figure 6A). Besides, we added the complete GO and KEGG results to the supplementary data (Table S10, Table S11).

Point 8: The conclusions section in this manuscript requires improvement to provide a clearer explanation of the study's results, its significance, and the implications for future research.

Response:Thank you for pointing that out, and it has been changed (See Page 12-13, the highlighted part)

Reviewer 3 Report

Comments and Suggestions for Authors

The authors performed massive sequencing of 96 DNA samples of Ujimqin sheep and made an association analysis with the number of vertebrae, in order to identify candidate regions or genes determining the "spinal development" or “number of vertebrae” trait. They identify the ABCD4 gene. The manuscript is well written, although the discussion might be improved.

Minor suggestions have been included in the manuscript .pdf file.

Author Response

Reviewer: 3

Comments and Suggestions for Authors

The authors performed massive sequencing of 96 DNA samples of Ujimqin sheep and made an association analysis with the number of vertebrae, in order to identify candidate regions or genes determining the "spinal development" or “number of vertebrae” trait. They identify the ABCD4 gene. The manuscript is well written, although the discussion might be improved.

Minor suggestions have been included in the manuscript .pdf file.

Point 1:  Page 2, Through whole-genome resequencing, we screened potential major genes associated with the variation in vertebrae number.” The aim of the research should be better explained, as many experiments are reported within the present manuscript.

Response:Thank you for pointing that out. We have changed the relevant content (See Page 2, the highlighted part).

Point 2: Page 5, These results provide evidence that the selective breeding of rams with multi-vertebrae traits is advantageous for the production of multi-vertebrae sheep and the establishment of a core population of multi-vertebrae Ujimqin sheep.” This sentence should be part of the discussion section.

Response: Thank you for your valuable advice, we have added it to the discussion section (See Page 11, the highlighted part)

Point 3: Page 6, The number of SNPs identified was similar to that reported in a previous study [27], suggesting that the sequencing data possessed high quality and could be utilized for subsequent analyses.” This is discussion.

Response: Thank you again for your constructive suggestions. We have added it to the discussion section (See Page 12, the highlighted part).

Point 4: Page 7, (Chr6: 79.566288Mb-81.566373Mb)” Did you check whether the SNPs were already published?

Response:Thank you for questioning, In the SNPs of top3, the SNPs on chromosome 6 do not annotate the gene (located in the intergenic region), so we check whether there are genes annotated by the GFF file (https://ftp.ensembl.org/pub/release-111/gff3/ovis_aries_rambouillet/Ovis_aries_rambouillet.Oar_rambouillet_v1.0.111.gff3.gz) in the upstream and downstream 1Mb regions (Chr6: 79.566288Mb-81.566373Mb) annotated by the GFF file of the two SNPs. Since there are no other genes annotated in this region, the sites in this region are not of our concern, so we do not know whether the SNPs in this region has been published. However, what we do know is that the two SNPs we are concerned about, SNP1(Chr7:89387652, G>A) and SNP3 (Chr7:89393414, C>T), have not been published yet.

Point 5: Page 9, The results showed that SNP2 (Chr7:89384139, G>C) was not observed in the sequenced individuals.” how do you explain this?

Response:Thank you for questioning again. “SNP2 (Chr7:89384139, G>C) was not observed

in the sequenced individuals.” means that in our subsequent sequencing results, we found that there was no base mutation from G to C (the opposite chain in the Figure 5). In fact, this is normal, because this site is not the significant SNP we obtained by GWAS analysis (significant SNP is SNP1), it is one of the two missense mutations we found after annotating the ABCD4 gene. Our sequencing is to verify that the mutation at this site, like SNP1 and SNP3, occurs only in multi-spine sheep. It turns out that there is no mutation at this site in the sanger sequencing results of multi-vertebra sheep, suggesting that this site may not be a mutation that affects the variation of the number of vertebrae in sheep. In addition, we retain SNPs with maf > 0.05 for analysis(The SNP will be recorded as long as it appears in more than 5 individuals), so not all loci will mutate in every individual. SNP2 is likely to exist only in a small number of sheep individuals and has nothing to do with spine number variation.
